# The Pathological Activation of Microglia Is Modulated by Sexually Dimorphic Pathways

**DOI:** 10.3390/ijms24054739

**Published:** 2023-03-01

**Authors:** Jennifer L. O’Connor, Jillian C. Nissen

**Affiliations:** Department of Biological Sciences, SUNY College at Old Westbury, Old Westbury, NY 11568, USA

**Keywords:** microglia, sexual dimorphism, inflammation, autoimmunity, glioblastoma, infection

## Abstract

Microglia are the primary immunocompetent cells of the central nervous system (CNS). Their ability to survey, assess and respond to perturbations in their local environment is critical in their role of maintaining CNS homeostasis in health and disease. Microglia also have the capability of functioning in a heterogeneous manner depending on the nature of their local cues, as they can become activated on a spectrum from pro-inflammatory neurotoxic responses to anti-inflammatory protective responses. This review seeks to define the developmental and environmental cues that support microglial polarization towards these phenotypes, as well as discuss sexually dimorphic factors that can influence this process. Further, we describe a variety of CNS disorders including autoimmune disease, infection, and cancer that demonstrate disparities in disease severity or diagnosis rates between males and females, and posit that microglial sexual dimorphism underlies these differences. Understanding the mechanism behind differential CNS disease outcomes between men and women is crucial in the development of more effective targeted therapies.

## 1. Introduction

Microglia play a key position in the pathology of essentially all brain diseases, in-cluding but not limited to the onset and progression of neuroinflammation, autoimmune dysfunction, tumor proliferation, and neurodevelopmental or psychiatric disorders [1,2,3,4]. However, the fundamental nature of these pathologies varies widely, as some are characterized by excessive inflammatory immune activation while others progress through immunosuppression. This disparity can be explained by the fact that microglial cells, upon encountering a perturbation of their environment, can assume a functional phenotype that falls upon a spectrum of activation states. In some scenarios, microglia can act in an inflammatory manner whereby they inhibit proliferation of nearby cells and stimulate immune system activity but also can cause adverse tissue damage outside of their clearance of cellular debris [5]. On the other hand, microglia can promote cell proliferation and tissue repair; producing effector molecules relevant to an anti-inflammatory role that can function in a tumor-supportive manner [6]. Further, temporal control of microglial-mediated inflammation is critical in maintaining CNS homeostasis, as acute activation allows for injury or pathogen invasion to be resolved with minimal off-target damage, while prolonged and over-exuberant cellular activation can result in self-sustained damage often leading to neurotoxic phenotypes, chronic inflammation, and related diseases [7]. In the present review, we discuss environmental and developmental factors which trigger microglial activation states to polarize into pro-inflammatory or anti-inflammatory phenotypes in pathological contexts, particularly focusing on sexually dimorphic regulation of their activation state. In this context, we provide a comprehensive analysis of their clinical presentations as a mechanism that underlies variations in disease susceptibility and outcomes between males and females.

## 2. Historical Definition of Microglia

Neuroglia (glia) were first defined in 1856 as a distinct brain cell population dissimilar to neurons by Rudolf Virchow [8]. Virchow’s postulations largely inaugurated the similar, yet slightly different conceptualization of glia as a connective network. The goal to refine the distinction between brain cell types proceeded through the 19th century, yet did not gain any clear significance until 1895, under the observations of Carl Weigert [9]. Weigert’s argument held the general idea regarding neuroglia as a somewhat matrix-like lattice containing dispersed nuclei throughout. Further distinction of glial cell types slowed in 1919, following the introduction of today’s accepted terminology for the separation of microglia, astrocytes, and oligodendrocytes as distinctive glial cell-subtypes [10].

However, a slight misconception arguably still resides within the boundaries of the naming of microglia themselves. As rooted in its Greek origin, the prefixes of micro- and glia- literally spell out “small-glue”. Ironically, microglia are not attached (or glued) to any other cells, and instead are mobile in their ability to scan their local microenvironment. Microglial involvement with a variety of processes including tissue homeostasis places them in a critical role in the maintenance of normal CNS function [11,12]. In the current view, microglial cells constantly sample the interstitial fluid as immunocompetent cells and constitute 5–10% of the brain’s overall cellular landscape [13,14].

The immune system has both adaptive and innate components; however, the CNS is considerably immune-privileged by its blood–brain barrier (BBB). The BBB, which is made up of brain microvascular capillary endothelial cells, projections of astrocytes, neuroglial membranes, and podocytes, functions as an innately shielding filter from toxic or infectious substances in the peripheral bloodstream, whilst supplying essential nutrients to the brain [15]. As functional BBB-residents, triggering of the neuroimmune interface inherently relies on the functionality of microglia as they rapidly respond in any context to minimize neural damage.

## 3. Developmental Origin of Microglia

Preconceived notions postulated the early embryogenesis and differentiation process of microglial cells originated from hematopoietic stem cell origins [16]. Today’s researchers have unveiled that microglia are of non-mesodermal origin and mediate their own auto-proliferation into the adult years [17,18]. In early embryogenesis, yolk-sac-originating progenitor microglia invade the neural tube and disperse throughout the nerve tissues and brain mesenchyme [19]. These progenitors plant themselves within the brain rudiment and remain present into adulthood [17]. Microglial colonization of the CNS occurs well before the formation of astrocytes and oligodendrocytes, which are derived from the neuroectoderm prior to hematopoiesis [20]. However, accurate measure of the true numbers of neuroglial cells has been a challenging target that is the subject of prolonged debate [21]. During early developmental stages, microglia disseminate throughout the CNS in a somewhat homogeneous manner whilst simultaneously undergoing an adaptive determination of their phenotype. Subsequent plastic changes are mediated by several factors, including regional localization of cells, internal and external environmental cues in the host, and sexually dimorphic developmental regulation [22,23] As residents of the CNS, microglia experientially assume their roles over time, thereby maturing into their functional capacities.

Recent studies have demonstrated several novel and fundamental roles for microglia, such as in the development of the neuronal network via modulation of neuronal proliferation and differentiation [23,24,25], the formation of synaptic connections and microglial-synapse interactions [25,26], their interaction with neuronal precursor cells throughout the lifetime of the organism [27], involvement in neuroprotection and neurotoxicity [28], and priming of neurodegenerative senescence [29]. Studies particularly within the last decade involving genetic engineering of new EGFP-reporting transgenic rodent models have provided researchers with a more clear-cut method to study the many dynamic processes of microglia in vivo [30]. Surgical implantation of a thin cranial window plate in mice for two-photon fluorescent microscopy has allowed a number of pioneer neurobiologists to influence the birth of an arguably new outlook regarding the overall purpose of microglia [31]. The implementation of improved experimental methods has provided a plethora of new perspectives within diverse physiological contexts. Recent studies have revealed some of the double-edged sword-like qualities of microglia, particularly related to their inflammatory responses to diseases.

## 4. Microglia Are Phagocytes of the CNS

Microglia and macrophages are often referenced as analogous in certain respects, mainly relevant to the similarities of their phagocytic-, cytotoxic-, and endocytic-involved processes to destroy foreign materials [14,32,33]. The two cell types also share similarities in their antigen-presentation strategies, as observed in their myeloid-like receptors for chemokines, cytokines, and pattern-recognition receptors [34,35]. These cells diverge when observing their physiological niche. Microglia live primarily within the brain’s parenchymal regions, and over the course of one’s lifetime undergo constant processes of self-renewal and replacement without any aid from monocytic precursors in the periphery [36]. Peripheral macrophages or CNS-associated macrophages, in contrast, are restricted within the CNS to the peripheral overlapping meningeal membranes, choroid plexus, and perivascular spaces in a typical healthy subject [37].

Microglia and macrophages both demonstrate several mechanisms to maintain homeostasis in the context of inflammation with the help of their antigen presentation strategies [38]. These two cell types share many similarities, such as in relevant pattern recognition receptors for damage-associated molecular patterns (DAMPs) or pathogen-associated molecular patterns (PAMPs). Common examples of PAMPs are derived from the detection of lipopolysaccharides, endotoxins, dsRNA, or flagellin, among others. PAMP receptors are typically classified in four categories: toll-like receptors (TLRs); RIG-I-like receptors; lectin-type, β-glucan, and mannose receptors; and NOD-like receptors [39]. Additional microglial receptor complexes include triggering receptor expressed on myeloid cells (TREMs), CD36, and scavenger receptors such as scavenger receptor class B member 1 (Scarb1) or macrophage scavenger receptor 1 (MSR1). The expression of certain cellular markers, intracellular and surface proteins, and other molecule secretions makes the distinction between microglia and macrophages quite complex because the two cell types contain several shared markers such as F4/80, Iba1, CD45, CD11b, and Cx3R1 [40].

A notable difference between the two cell types involves the expression of the CD44 marker found in macrophages yet not in microglia [41]. Additional transcriptome studies have suggested the absence of CD169 markers in adult microglia in comparison [42]. Further research has drawn particular attention to the phagocytosis of apoptotic cells, also known as efferocytosis, as microglia have been posited to show more efficient facilitation of this process [43]. Efferocytosis is proposed to minimize damage from surrounding cells. Phagocytic cells engulf apoptotic cells; for microglia this is particularly relevant in early developmental stages as well as over biological senescence/aging [44]. In addition, in the observation of the process of cell death known as primary phagocytosis or “phagoptosis,” it has been shown that inflammatory activation of microglia factors into pathogenic models of neurodegeneration due to the loss of a non-engulfment signal or sudden exposure to an allowed-engulfment signal on viable neurons [45,46,47]. In vitro and in vivo models have revealed microglia as coordinated acute neuroinflammatory active responders promoting phagocytosis of damaged neurons and toxic aggregates, while also attempting to refurbish CNS tissue to maintain relative homeostasis [48,49].

Microglia phagocytose, engulf, and clear β-amyloid and α-synuclein in neurodegenerative models of Alzheimer’s and Parkinson’s disease, and have been reported to mechanistically rely on receptors such as TLR4 or β1 integrin, suggesting this process as neuroprotective against degeneration. However, prolonged phagocytosis can contribute to neurotoxic buildup and result in over phagocytosis of neurons, reported in a neurodegenerative model of synapse opsonization through the complement system [50,51]. Ideally, competent microglial phagocytosis requires the cooperation of a variety of scavenger receptors and inflammatory-signal recognition receptors [52]. In addition, microglia express receptors for misfolded proteins or apoptotic cellular debris such as scavenger receptors, galectin 3, mannose receptor, pyrimidinergic receptor P2Y6, and TREM2 [53,54,55,56,57]. Furthermore, the facilitation of phagocytic activity has shown to release acute-phase proteins, whereby their expression is prompted by respective TLR ligands [58]. Collectively, receptor-mediated interactions organize the microglial actin cytoskeleton and assist in eliminating harmful foreign substances, while disruptive modifications to microglial phagocytic capacity contribute to the onset of several CNS diseases.

## 5. Microglial Roles in Neuronal Architecture

Microglia are professional phagocytes of the brain, yet they are also undoubtedly crucial in early brain development due to their modification of the turnover of neuronal circuitry in the brain as it establishes sufficient neuronal highways over time [59,60]. These cells have the capacity to regulate or eliminate synapses and cellular substructures, to the extent that about half of the neurons generated during neurodevelopment are ultimately discarded [60]. Microglia have also demonstrated a role in neurogenesis, migration of interneurons, and in synaptogenesis via the astrocytic secretion of cholesterol and other microglial trophic factors [61,62,63,64]. In addition, microglia are also capable of reducing synaptic adhesion molecules responsible for stabilizing a synaptic connection such as N-cadherin, protocadherin, and SynCAM1 [65]. Modification and resultant efficacy of synaptic connections are therefore enhanced and refined thanks to microglia. Although the formation of synapses involves several other cellular components to specialize neuronal function, microglia are assumed to be a key regulator in the selective remodeling of synapses by presynaptic trogocytosis and spine head filopodia induction for axonal pruning regulation by the complement system [24,66].

Without microglia, neurons fire more frequent action potentials due to the lack of the selective removal of excess or silent synapses, apparent in brain slice culture modeling the inhibition of synaptic pruning following microglial ablation [67]. Complex signals such as C1q and C3 complement factors have also shown a role in tagging synapses for elimination alongside microglia and neighboring astrocytes [68]. As integral cytoarchitectural elements and modulators of synaptic transmission, microglia provide the framework for neuronal plasticity throughout the course of adulthood [69]. Microglia preferentially remove less active synaptic inputs, as neuronal activity regulates the signals responsible for their engulfment [69]. They have also revealed a mild capacity to displace synapses in the synaptic refinement process known as stripping, resulting in activation-mediated morphological changes [70,71]. Researchers have observed that microglial encapsulation of neuronal cell bodies and subsequent synaptic displacement promotes neuronal activity; stimulation of their NMDA receptors promotes calcium influx into neighboring neurons. Further, these neurons then release neuroprotective factors that are involved in the prevention of apoptosis and in the support of neuronal survival [72]. Thus, the developmental regulation, tagging of synapses at critical periods, and preferential engulfment of axons is physiologically regulated by microglia. Current research efforts focus on the continuous negative feedback control of neuronal activity by these resident immune cell populations. One potential strategy to view the sculpted effects left behind due to the experiential microglial influence on neurons is the development of region-specific microglial ablation approaches. A more mechanistically refined understanding of the localized effects of microglia on neuronal activity intends to encompass their tendency to preferentially balance the activation and latter retraction of their processes in various experiential contexts. The pathological presence or absence of microglia has emphasized their role in the maintenance of a homeostatic environment within the CNS and the formation of healthy and functional neural routes over time.

## 6. Microglial Activation States

Under normal, healthy conditions, microglia are quiescent and lack endocytic or phagocytic activity. These cells constantly scan their environment and are able to survey the whole brain in a few hours [22]. For example, the scanning behavior of microglia can be impacted by their triggered activation. Researchers have reported an increase in the size of the territory scanned by the processes of a single microglial cell and an increase in bidirectional velocity upon its recognition of acute noxious stimuli [73]. Their branches host extensive radial processes, appearing as cytoplasmic protrusions and projections [74]. The immune response microglia encompass is generally innate, in that their response to a problem area is instantaneous and could be considered a non-specific approach. Upon recognition of an inflammatory insult, microglia transform from a sessile to an activated state and secrete a variety of inflammatory factors, leading to the shortening of their external branches and subsequent retraction of their ramified processes [75] In the progression of their activation, microglia can convert all the way to a macrophage-like state to actually engulf and destroy pathogens [76]. In common and less complex cases following their activation, microglia return back to their sessile state and continue their scavenging of the CNS. However, in chronic inflammatory states microglia remain present and continue to clear debris, eventually damaging normal functioning cells leading to long-term destruction of neuronal tissue, atrophy, and neurodegradation [77].

It is important to note that microglia do not assume a homogeneous functional profile upon activation, but rather assume a wide spectrum of activation states. Historically, microglia were considered to be activated in a polar manner between M1 and M2 phenotypes. M1 microglia were generally pro-inflammatory and neurodegenerative, while M2 microglia were anti-inflammatory and healing. In this schema, microglia that have undergone classical M1 activation produce pro-inflammatory cytokines and chemokines, such as interleukin (IL)-6, IL-12, IL-1β, and tumor necrosis factor alpha (TNFα) [77]. These microglia also express major histocompatibility complex-II (MHC II), integrins such as CD11b and CD11c, the costimulatory molecules CD45 and CD47, and Fc receptors, which all can contribute to neural damage [78]. In addition, these cells express NADPH oxidase and are known to produce inducible nitric oxide synthase (iNOS) with subsequent nitric oxide (NO) release, among other reactive oxygen species (ROS) [79]. On the other hand, in this framework alternative M2 activation is induced by anti-inflammatory cytokines such as IL-4 and IL-13, and M2 microglia secrete anti-inflammatory cytokines such as transforming growth factor beta (TGFβ) and IL-10, as well as the Th2 recruiting factor CC chemokine ligand 2 (CCL2) [80,81]. M2 microglia also produce fibroblast growth factor (FGF), insulin-like growth factor-1 (IGF-1), colony stimulating factor-1 (CSF-1) and neurotrophic growth factors such as glial-derived neurotrophic factor (GDNF1), brain-derived neurotrophic factor (BDNF), nerve-derived growth factor (NGF), and pro-survival factor granulin (PSFG) [81]. However, recent studies have argued that a polar model of M1 and M2 phenotypes is overly reductive of the wide range of microglial functions [82]. Further subdivisions have been made to the M1-M2 framework with the inclusion of M2a, M2b, and M2c subtypes [83]. Despite many studies exploring the nature of this issue, there is no updated unified standard for delineating microglial activation states. Thus, even though the M1/M2 terminology has notable limitations in capturing the totality of microglial function, it is still widely used in recent literature. To this end, we will use the historical classification of M1 microglia to refer broadly to pro-inflammatory functionality and M2 microglia to denote generally anti-inflammatory functionality for the remainder of this review.

Studies have also shown that microglia have a neuromodulatory role, with the purpose of regulating homeostasis in the CNS. Microglia polarization may be further manipulated by the expression of a number of receptors for neurotransmitters, including several subtypes of metabotropic and ionotropic receptors for acetylcholine, adenosine, adenosine triphosphate (ATP), glutamate, GABA, adrenaline, noradrenaline, histamine, and serotonin [84,85,86]. Metabotropic receptors are expressed on the surface and coupled with second messenger associated intracellular signaling systems (i.e., Ca^2+^ signaling in microglial activation), whereas their ionotropic receptors involve the generation of ion fluxes. Outside of their possession of a variety of immune and phagocytic receptors for cytokines and chemokines, microglia have also shown to express P2X7 purinoceptors, and several tissue intermediaries including histamine, thrombin, bradykinin, and platelet activating factor [39]. To maintain physiological homeostasis in resident CNS tissue, microglia heterogeneously shift phenotypes based on experiential and environmental mediated factors such as second messenger cellular excitability and low resting membrane conductance [39].

## 7. Microglia Assume Diverse Morphologies in Disease or Injury

Microglia demonstrate abundant phenotypic plasticity in their morphology, expression of molecular markers, and distribution throughout the brain; because of this, many cellular marker-based analysis methods place limiting boundaries upon the attempt to define microglial cell types and morphological states. Fortunately, a combination of novel imaging and spatial statistic techniques, alongside transcriptome studies, have further quantitatively and qualitatively defined their function [87]. A five-parameter automated analysis algorithm of an Iba1 immunostain of retinal microglia in a murine optic nerve injury model used measures of cell density, nearest neighbor distance, and regularity index to gather histological data on microglia numbers and distribution, as well as cell soma size and roundness, noting that following injury microglia shifted from a small, round morphology to adopt a bigger, more irregular soma shape [87]. With a diverse range of states between ramified and ameboid microglia, correlating phenotype and function is quite the quarrel; however, recent studies have unveiled intermediate activation states and variants observed in several disease or injury pathologies such as hypertrophic, dystrophic, rod-like, rod-shaped, and DAM-activated [87,88].

Dystrophic microglia are the main disease associated morphology and have been noted as major factors in the progression of neurodegenerative disorders [89]. Dystrophic microglia have been proposed as unrelated to age or senescent neurodegeneration, as opposed to hypertrophic microglia, which can produce chronic inflammatory mediators associated with neuroinflammation and inflammatory aging [89]. In a study of an aphasic variant of Alzheimer’s Disease (AD), hypertrophic microglial densities were positively associated with ramified microglia densities, along with the accumulation of neurofibrillary tangles [90]. These hypertrophic microglia displayed a higher immunoreactivity for a marker of the Human Leukocyte Antigen—DR isotype (HLA-DR) and peptide complex of MHC II receptors in their inflated cell bodies and both thicker and shorter processes, while ramified microglia displayed smaller cell bodies and more intricate branching [90]. Visually, dystrophic microglia appear disjointed with bead-like and fragmented branches with sparingly thin channels connecting the seemingly separate portions. Hypertrophic microglia have shown to express the senescent marker P16^INK4a^ and surround amyloid plaques in tauopathy models [91]. Furthermore, the upregulation of the P16^INK4a^marker by these microglia has demonstrated their capacity to target the removal of senescent cells, resulting in reduced tau aggregation pathology, neurodegeneration, and cognitive impairment [92].

Current research also accounts for additional activation variants in the diseased, injured, and aging brain [93]. Rod-shaped and fully formed rod microglia are similar in that they appear elongated, cylindrical, contain a shrunken soma, and retracted side processes [94]. These microglial subtypes are often observed near neuronal elements that are damaged or are vulnerable to damage. Observations of fully formed rod microglia note their planar processes often project from the apical and basal ends of the cell. Non-specific antigen staining methods complicate the distinction between variant rod-shaped and rod microglia, although a possible difference may reside within the rod microglial tendency to condense at the compact spaces in degenerative white matter tracts, independent from pathological gray matter [95].

Transcriptional single-cell sorting in rodent models has also elucidated a conceptually different novel sub-population of microglia, known as disease-associated microglia (DAM). DAMs are recognized in mouse models of AD, where researchers have reported a downregulation of *AIF1* and homeostatic genes such as *Cx3Cr1*, *P2RY12*, and *TMEM119*, and an upregulation of *CD68* and *CD74* [96]. Reports of DAM activation states are notably mediated by both TREM2 signaling and binding to apolipoprotein (APOE) [97]. TREM2 is present on homeostatic microglia, and senses phosphatidylserine on damaged and apoptotic cells to activate APOE signaling and induce an autonomous microglial neurodegenerative phenotype (MgND) [98]. MgND microglia further induce inflammation by the secretion of molecules such as iNOS, NOS, secreted phosphoprotein 1 (SPP1), macrophage colony-stimulating factor 1 (mCSF1), CLEC7A, and miR-155 [98]. Recent evidence demonstrates a link between the deficiency of TREM-2 in vitro and a decrease Aβ uptake by APOE-4 treated microglia [99]. DAM subpopulation-involved pathology is apparent in AD models and is also reported in TREM2-APOE pathway models of amyotrophic lateral sclerosis (ALS) and multiple sclerosis (MS) [100,101].

## 8. Sexually Dimorphic Development of the Neuroimmune System

Differentiation of sex arises from the overlapping of mammalian chromosomal, gonadal, hormonal, phenotypical, psychological, and epigenetic factors. The process of sexual differentiation begins in early embryogenesis, where distinct gonadal tissue in the form of male testes (XY) or female ovaries (XX) promote a pivotal event for the translation of chromosomal sex into phenotypic sex. The SRY gene is believed to promote a genetic network which ultimately mediates the differentiation of primary (mediated by genes) and secondary (mediated by hormones) sex characteristics such as distinct hair patterns, musculoskeletal features, and organismal size differences [102]. Since females have two X chromosomes, two copies likely account for their heightened protection against X-linked hereditary diseases and general tendency to outlive males; however, individuals with XX chromosomes have a higher risk of autoimmune diseases [103].

In rodent models, fetal gonads are active by mid to late gestation, excluding spermatogenesis where androgen production by fetal testes occurs at the last few days of gestation and briefly post-birth [104]. In primates, the production of androgens occurs at the end of the first trimester, into a few weeks of the second trimester, and peak at birth [105]. Masculinized endpoints are induced by the hormone testosterone when it aromatizes into estradiol (E2), 5-α when it is reduced to dihydrotestosterone (DHT), or both [106]. Microglia are also sensitive to estrogen and testosterone after enzymatic conversion due to their physiological expression of receptors for steroid hormones; however, the organizational effects of these hormones differ across brain regions [107]. Brain feminization phenotypes have been shown to involve the prevention of masculinization by epigenetic regulation, whereas masculinization is tightly regulated by gonadal hormones.

The stabilization and formation of masculinized dendritic spines occurs after estrogen is aromatized within a neuron by the aromatase enzyme. After binding to the estrogen receptor, it translocates to the nucleus where it then transcribes the cyclooxygenase-1 (COX-1) and COX-2 genes. These genes convert arachidonic acid into a potent inflammatory mediator known as prostaglandin E2 (PGE2) [108]. Following the conversion, PGE2 is proposed to promote the phosphorylation of PKA following interaction with its EP2 and EP4 receptors. The further phosphorylation of PKA acts on GluR2 subunits of neuronal AMPA receptors, which may explain the phenotypic observation of their membrane clustering. Microglia can produce and respond to lipid prostaglandins and utilize these signals to respond to sites of inflammation [108]. Microglia also have been shown to secrete prostaglandins to trigger the calcium-dependent release of glutamate by neighboring astrocytes to ultimately stabilize and form dendritic spines [109,110,111,112]. Prostaglandins have further roles in the cross-talk between mast cells and microglia and are proposed to drive the process of sex differentiation, notably in the brain’s preoptic area (POA) [112]. Rodent model male microglia displayed a tendency to appear ameboid and activated, characterized by phagocytic cups, reduced process length and branching, and an enlarged soma, compared to female with a more ramified and surveying phenotype [112]. Outside of the chromosomal determinants of phenotypic sex, a single injection of PGE2 was observed to play a role in the masculinization of sexual and play behavior in rats consistent with the presence of more mast cells during the sensitive period and also across development in males [112].

Recent studies demonstrate sex differences between male and female rodents in various brain areas relating to density and phenotype [113,114,115,116]. The spatiotemporal distribution of microglial density varies across different developmental stages and displays recognizable regional heterogeneity. In early development, there is an elevated microglial density in the female hippocampus, cortex, cerebellum, and striatum [113]. Male microglial density is notably elevated in the amygdala during early developmental stages; however in early adulthood these cells are elevated in the cortex and hippocampus [114]. Furthermore, during development there have been reports of an enlarged soma in male microglia throughout the brain throughout adulthood, whereas female microglia have a larger soma in the cortex, hippocampus, and amygdala [113]. Rodent studies have demonstrated male rats at P4-days old have more ameboid microglia in the cortex, hippocampus, and amygdala, whereas at P30-days old females have more activated microglia in those regions, and at P60-days old females have more activated microglia [115]. Other studies have reported the longitudinal development of these areas and found C-C motif chemokine ligands such as CCL4 and CCL20 were increased 50- and 200-fold, in males compared to females, later colonizing in brain areas involved in memory and cognition [116]. Overall, it is important to note the amorphous role of microglia because they are constantly receiving signaling input from local, environmental, and experiential factors over the course of aging.

## 9. Sex Differentiated Microglial Function through Adulthood

An individual’s responses to inflammatory insults are iterative manifestations of several factors such as genetic predisposition, environmental stimuli, and exposure to disease-priming factors in utero such as their mother’s malnutrition, stress, infections, microbiome discrepancies, and insufficient caregiver interactions [117]. Cumulatively over time, these insults can synergistically act to promote the onset of several neurodevelopmental complications [118]. Divergences in the process of microglial maturation between males and females prime sexually dimorphic responses to perturbations of CNS homeostasis in the adult. An experimental microarray and qPCR analysis of microglial and inflammatory hippocampal gene expression of male versus female mouse samples from ages 3 months (young), 1 year (adolescent), to 2 years (old) were compared alongside the monitoring of the female estrous cycle stages until reproductive senescence had been established [119]. Age-regulated genes were analyzed and were enriched in microglia-specific transcripts, such as inflammation-related ligands (*C1qa*, *C1qc*, *CCL4*), as well as effector- and receptor-transcript encoding proteins for sensing endogenous ligands or microbes referred to as the microglial sensome [119]. In the early postnatal and juvenile period, male microglia were more sensitive to lipopolysaccharide (LPS) and showed an augmented microglial transcriptional maturation as opposed to females [119]. However, by early adulthood (day 60), male microglial development began to downregulate, and female microglia assumed a more developmentally mature phenotype from that point forward, as evidenced by analysis of gene expression patterns correlated to the microglial developmental index (MDI). Further, in comparison to age-matched males, female microglia demonstrated an increased expression of transcripts related to inflammation over the course of aging into adulthood, which is suggestive of amplified microglial activation over a female’s lifetime **[119].** These data were recapitulated by other groups, similarly showing increased MHC I, complement, and other inflammatory genes [120] as well as an increased inflammatory response to LPS [121] in aged female mice relative to males.

Sex differences have been reported regarding microglial migratory activity, which is higher in male microglia compared with that from females and exclusive from differential mRNA expression of the interferon gamma (IFNγ) receptor [122]. Increased motility is characteristic of anti-inflammatory, M2 microglial populations [123]. In addition, male microglia express more M2-associated P2Y12 purinoreceptors involved in chemotactic motility towards ADP and ATP gradients, and P2X_7_ receptors for extracellular ATP in early developmental stages [124,125]. An alternative study that focused on morphological sex differences in mice during the first week after birth reported microglia as demonstrative of an inflammatory-like morphology that converts to a largely ramified state by the third week postpartum and completion of male defemination [126].

Synthesis of estrogen related to the neonatal androgen surge from male testes may manipulate organizational patterns involved in determining a sexually categorized phenotype [107]. Steroids including but not limited to testosterone, allopregnanolone, and 17β-E2 appear to mediate microglial involvement in neuroinflammation, and in turn microglial neuroinflammation can impact the synthesis of neurosteroids [116,127]. Additionally, sex hormones such as testosterone can increase the production of anti-inflammatory IL-10 and provide protection against autoimmunity; however, the effects of such hormones in females change after puberty which raises autoimmunity risk and disease prevalence [128,129,130]. Estrogen has shown a role in the regulation of the immune response by inhibiting the effects of autoreactive B cells and enhancing the expression of inflammation-associated CCR5 markers [131,132,133]. Further, estrogen itself has been shown to have a sexually dimorphic effect, in that E2 administration promoted an anti-inflammatory effect in male microglia and conversely a pro-inflammatory effect in female-derived cells (Figure 1) [134]. The exact mechanism of these hormone-induced changes is elusive, yet neonatal treatment of the female brain with estrogen in a brain defemination protocol suggests an altered expression of genes occurs in early embryogenesis [135]. The process of microglial sex determination may begin prior to hematogenesis, evident by estrogen-induced changes promoting an enhanced response to inflammatory stimuli in males that are also in agreement with microglia transcriptome data [136]. In the early postnatal period, there is increased expression of the inflammatory cytokines TNFα and IL-1β in female mice compared to males [137]. Collectively, females and males demonstrate sex-specific immune responses relevant to several CNS contexts. Studies show sexual differentiation displays different functional outcomes in CNS injuries and diseases [136]. The male associated hormone testosterone substantially reduces the reactivity of microglia and astrocytes after brain injury, suggesting it serves an immunosuppressive role [126]. Researchers have previously reported an interaction between the hypothalamic-pituitary adrenal (HPA) axis and inflammatory responses to a stimulus (IL-6) in both intact and orchiectomized rats, in which stimulation of hormone release, such as corticosterone and adrenocorticotropic hormone (ACTH), was mitigated by testosterone replacement therapy (TRT) [138]. Higher testosterone levels were highly associated with the downregulation of all cytokines [139].

Neuroinflammatory-induced changes between males versus females differ in magnitude and dynamic functionality over the course of an individual’s lifetime. Interpretations from clinical studies demonstrate testosterone suppressed inflammation in patients with diabetes mellitus, coronary heart disease, and prostate cancer, decreased the expression of pro-inflammatory cytokines (TNFα, IL-6, and IL-1β), and increased the expression of anti-inflammatory cytokines (such as IL-10) [129,140]. Collective research interpretations demonstrate that basal gene expression profiles vary between males and females over their lifetime, with an overall predominance of pro-inflammatory M1 responses in females and anti-inflammatory M2 responses in males, as summarized in Table 1. This dichotomy provides a plausible determinant of sexually dimorphic immunity, which is a critical mediator of CNS disease progression and prognosis.

## 10. Microglial Sexual Dimorphism as a Mediator of Divergent CNS Disorder Outcomes

Epidemiological research has overwhelmingly recognized a differential susceptibility, onset, progression, and outcome in several CNS injuries, neurological disorders, and neuroinflammatory diseases between males and females. For instance, at birth, males are more often born prematurely and are more likely to sustain a prolonged injury compared to their sisters, who may have the same exact injury yet a less pathologic outcome [141,142]. There is also a strong sex bias in the incidence of developmental neurological disorders such as attention deficit hyperactivity disorders (ADHD), autism spectrum disorders (ASD), dyslexia, stuttering, Tourette’s, and early-onset schizophrenia towards males [143,144,145,146]. Alternatively, female disease sex-biases tend to not occur until during or after puberty, and a heightened incidence of neurological disorders often appears post-menopause. These disorders include but are not limited to late onset schizophrenia, anorexia, bulimia, and multiple sclerosis (MS) [147,148,149].

In the remaining scope of this review, we turn our attention to prolonged neuroinflammatory diseases - autoimmune disorders, pathogenic CNS invasions, and glioblastoma multiforme, all of which demonstrate sex-differential CNS disease outcomes. Here, we mechanistically propose that microglial involvement in a host’s disease onset and progression is largely responsible for sexually dimorphic CNS pathologies. The active building of one’s immune and synaptic profile over their lifetime occurs in multifaceted trajectories that diverge between males and females, and we suggest that these cumulative experiences can profoundly impact how microglial cells modulate the CNS in health and disease.

## 11. Female Predominance in Autoimmune Disorders Is Mediated by M1 Microglia

Epidemiological studies of CNS autoimmune diseases have reported women are more frequently affected compared to men, particularly after the onset of puberty throughout adulthood [150]. Further clinical research insight has demonstrated female MS patients are three times more likely to be affected by relapsing-remitting multiple sclerosis (RRMS) and other secondary progression subtypes [151,152,153]. Areas of damage or scarring in the CNS of MS patients (lesions) are caused by inflammation or the attack of myelin sheath in the brain, spinal cord, or optic nerve and are pathologically heterogeneous [154,155,156]. Radiological studies have indicated women with MS have more severe inflammatory lesions compared to males [157]. Female patients with RRMS displayed a more profound neuroinflammatory activity, yet not in the neurodegenerative component of MS known as primary progressive MS (PPMS), which is equivalent in both sexes [158,159]. A large portion of men with MS have lower levels of testosterone circulation and an increase in disease-associated disabilities [160,161]. As discussed above, this may relate to how testosterone demonstrates some anti-inflammatory and neuroprotective properties, with lower levels correlating to increased risk for MS development [160].

Additional sexual dimorphism has been reported in the progression of Neuromyelitis Optica (NMO) spectrum disorders (NMOSD), characterized by symptomatic conditions clinically manifesting as prolonged optic neuritis, myelitis, and brainstem encephalitis. However, there have also been reports of at least six diverse lesion types and mechanisms of tissue injury [162,163,164]. Approximately 80–90% of patients experience a relapsing course of NMOSD due to the harboring of serum antibodies to the astrocyte water channel aquaporin-4 (AQP4); this is used as a diagnostic biomarker known to correlate to increased degrees of CNS tissue damage [165,166,167]. In a recent epidemiological study, females over the age of 40 were the most susceptible and most likely to test positive for AQP-4 antibodies compared to males and other age groups [168]. These disparities are summarized in Table 2.

As discussed above, the immune response is sex-differential, and may hold substantial influence on the differential onset of autoimmune disease phenotypes. Studies have often reported the role of astrocytes in complement activation; however, emerging roles of microglia have implicated they largely modulate the pathogenesis of multiple sclerosis (MS) and neuromyelitis optica (NMO) [174,175]. Analysis of experimental autoimmune encephalitis (EAE) models for data regarding sex-differential immune cell profiles, BBB crosstalk and peripheral trafficking approaches may provide more insight into the molecular mechanisms driving sexually dimorphic disease traits addressed in Table 3. The chronic activation of microglial M1 phenotypes in women may serve as potential cofactors in the female-biased diagnoses of RRMS and NMOSD.

### 11.1. Multiple Sclerosis and Mechanisms of Microglial Involvement

MS is an autoimmune disease where the immune system attacks and damages the myelin sheath on surrounding neurons, nerve fibers, and oligodendrocytes which normally function in the production of myelin. Microglial activation, subsequent oxidative injury, and mitochondrial damage have shown to trigger histotoxic hypoxia and energy deficiency processes to pathologically promote the onset of demyelination and neurodegeneration in the progressive stage of the disease [176,177]. Destruction of CNS myelin is commonly associated with the activation and crosstalk of microglia and macrophages [175]. In the event of active MS lesions, astrocytes undergo reactive protoplasmic or fibrillary gliosis, and in patients with aggressive MS, loss of AQP4 has been observed in early MS stages. Infiltrating T lymphocytes, B lymphocytes and macrophages promote intense microglial activation states in periplaque white matter and promote an attraction towards perivascular cuffs or dispersal within the vicinity of local parenchymal lesions [164].

Microglia tend to demonstrate an immediate classical M1 phenotype in the early onset and progression of MS, alongside an expression of proinflammatory cytokine macrophage inflammatory protein (MIP) [178,179,180]. Pharmacological ablation of microglia or polarization to an M2 phenotype substantially reduces EAE symptoms [181,182,183,184]. MS CNS lesions cross the BBB and are characterized by inflammation, demyelination, gliosis, and external neurodegeneration, leading to several failures or disruptions of proper neuronal signaling [179]. The exact mechanisms for MS and EAE are not fully clear; however, induction of rodent EAE in research shows pro-inflammatory molecules activate microglia-induced inflammation and leukocyte recruitment [185].

MS progression may occur with initial myelin-autoreactive CD4+ T helper cell activation in either the periphery or in the CNS, evident from studies of antigen-presenting B cell therapy in relapsing MS cases [180]. Activated CD4+ autoimmune effector T cells can increase the immune response by recruiting microglia and other immune cells after their crossing of the BBB [186]. CD4+ T cells reactivate via their interaction with antigen presenting cells, and subsequently secrete the inflammatory cytokines lymphotoxin and TNFα. These cytokines in excess are neurotoxic via an overactivation of microglia and astrocytes, amplifying local inflammation and damaging oligodendrocytes thereafter. Microglia can further induce myelin phagocytosis and cause increased reactivation of CD4+ cells, particularly in the close proximity of areas with prominent axonal loss and inflammation [187]. Current studies propose meningeal B-cell rich lymphoid aggregates may factor into MS by injuring underlying cortical neurons and triggering microglial activation [188]. Autoantibodies and complement factors also have observable effects on the CNS, further prolonging inflammation leading to neuronal damage [189,190].

Microglial and T cell interplay mediates neuroinflammation by a process involving myelin-derived antigen-presenting cell restimulation evident in a cocultivation of microglia with activated T cells in vitro [189]. Transference of T_H_17 and T_H_1 cells has demonstrated significant non-canonical nuclear factor kB (NF-kB) activation in rodent models [189,190]. Induction of microglia-specific NF-kB-inducing kinase (NIK) deficiency suggests a mechanistic microglial reliance on the NF-kB pathway [189]. Following CNS infiltration, T cells stimulate a microglial coordination of the noncanonical NF-kB signaling pathway with T cell-derived cytokines, such as granulocyte-macrophage colony-stimulating factor (GM-CSF), to induce expression of chemokines supporting an additional recruitment surge of T cells relevant to disease progression [191]. Conditional deletion of the NIK-encoding gene *Map3k1* suggests non-canonical NIK functions with microglia to regulate the flood of CNS-infiltrate T cells in late-phase EAE pathogenesis, yet interestingly not in early phases [189]. The NF-kB pathway has shown to encourage pathological functions of microglia in several other CNS pathologies [192,193]. Several genomic and pharmacological studies have focused on serine and threonine protein kinase signaling pathways such as MAPK in microglia, astrocytes, myeloid cells, T cells, and dendritic cells (DCs). These pathways are proposed to influence EAE pathogenesis, yet the refined understanding of these mechanistic process awaits further elucidation [194].

Lack of a stable and homeostatic microglial cell population correlates with disease and lesion progression in EAE rodent studies [195]. Macrophages and activated microglia revealed predominantly pro-inflammatory features in samples from MS patients [196]. Pro-inflammatory microglia remain linked with the reactivation of T cells and have shown to promote demyelination through their increased expression of MHC II complexes and secretion of cytokines and neurotoxic molecules [197]. Mice lacking the M1-associated CC-chemokine receptor 2 (CCR2) also showed resistance to EAE induction through the absence of monocyte infiltration into the CNS due to reduced antigen presentation leading to T cell activation. As portrayed in more aggressive forms of MS, increased oligodendrocyte and neuronal death in the cortex is promoted by microglial activation, however, autoreactivity may be further prompted by B-lymphocyte-forming lesions and ectopic lymphoid follicle-like aggregates within the meninges, of which may contain follicular DCs and T cells [198]. Taken together, today’s current data demonstrate the heterogeneity of immune cell cross talk over the BBB and within the CNS. A common thread between all these works is that MS pathology is mediated by inflammatory microglial cells that promote degeneration and loss of function in neurons, as summarized in Table 3. Overall, the predominance of M1 microglial phenotypes in females is a compelling mechanism to explain the drastic disparity in disease incidence between men and women.

### 11.2. Neuromyelitis Optica Spectrum Disorders and Mechanisms of Microglial Involvement

NMO is an autoimmune inflammatory disease resulting in the destruction of myelin sheath, particularly in the optic nerves and spinal cord. NMOSD is a term used to encompass optic neuritis and myelitis, as well as other diverse phenotypes and lesions. Astrocytes are proposed as the target for autoimmune disruption in NMO, as opposed to MS where oligodendrocytes are the focus of immune and inflammatory attacks [199,200]. Astrocytic cells are induced to produce large amounts of inflammatory cytokines, chemokines, and complement proteins through binding of an aquaporin-4 (AQP4) autoantibody, which results in local neurodegeneration. NMOSD is divided into seropositive and seronegative antibody expression forms, alongside the detection of AQP4-Ig, and/or the presence of myelin oligodendrocyte glycoprotein antibodies (MOG)-IgG [201]. Additionally, brain tissue distribution of activated microglia is dissimilar between MS and NMO, as in MS meningeal inflammation is generally more widespread and linked to cortical pathology with NMO legions centralized around the astrocytic end-feet at the BBB [202,203]. NMOSD lesions are typically located within areas such as the optic nerves, hypothalamus, and the diencephalic and brainstem tissues surrounding the cerebral aqueduct and third and fourth ventricles [204].

Positive AQP4-IgG NMO is generally associated with regions rich in AQP4 channel density such as hypothalamic and periventricular areas, as well as the corpus callosum [204]. Research data imply AQP4 internalization and decreased endogenous AQP4 expression as leading factors in the promotion of dysfunction in normal astrocyte foot processes along the BBB [205]. NMO-IgG is proposed as a priming factor in the coordination of adaptive immune processes promoting a variety of pathogenic outcomes [206]. Diagnoses of NMOSD are also positively associated with the expression of the pro-inflammatory cytokine IL-6. This subsequently binds to its receptor on microglia and triggers glycoprotein receptor 130 (gp130) subunit homodimerization, capable of initiating cellular action of the IL-6 family of cytokines [207]. Microglial activation in NMO disease models results in their increased gene expression for proinflammatory cytokines such as TNFα, IL-1α, IL-1β, iNOS, and activation of the M1-associated STAT3 signaling pathway [208,209,210] Microglia also respond to increased STAT1 and other inflammatory molecules such as type I interferon (IFN-1), which are then proposed to promote the expression of MHC I and II, CD86, and 2’-5’-oligoadenylate synthase ubiquitin (OAS) [209,210].

Complement convertase C3 has also demonstrated an involvement in the destruction of astrocytes and triggering of the production of C3a, which functions as a microglial-focal chemoattractant molecule towards astrocytes [211]. Researchers observed that activated astrocytic production of C3a and presentation of IgG antibody promote further NMOSD onset [212]. C3a binds to its receptor (C3aR) on microglia, thereby triggering their activation and the production of the classical cascade component 1 (C1q), resulting in the prolonged tissue damage seen in NMOSD patients [212,213]. C1q is primarily produced by microglia and C3 by astrocytes [212]. Microglial and astrocytic somas have demonstrated a tendency to overlap after AQP4-IgG infusion, further suggesting their interaction in NMO [199]. In two separate NMO rodent studies, astrocytes have shown to react to AQP4-IgG binding and a loss of AQP4 [199,214]. Loss of astrocytic AQP4 characteristically distinguishes NMO lesions from MS [215,216]. AQP4-IgG activated astrocytes are proposed to further promote local inflammation evident in NMO rodent studies and moreover confirmed by consistent human NMO patient data [217]. In addition, AQP4-IgG does not activate microglia directly; rather, astrocytic signaling induced microglial activation [218]. Although complement C3 is often absent in astrocytes in normal physiological conditions, in pathological conditions C3 is upregulated, and more specifically in NMO AQP4-IgG infusion upregulated astrocytic C3 [218]. The cleavage product C3a can function as a chemoattractant and may further induce astrocyte-microglial activation mediated by C3-C3aR signaling [218]. Following NMO-IgG administration, microglial activation was detected in lesion sites in concert with motor deficits in mouse models of disease. However, mobility was substantially improved following microglial ablation, indicating that microglia are critical mediators of disease pathology [199].

The exact role of complement when reflecting on microglial reactivity has yet to be determined. Varying serum concentrations of complement from several studies of NMOSD patients have shown significant microglial activation [219,220,221,222]. Similarly to MS, NMO can enter a remission phase, which leads to more questions regarding the role of microglia. To date, microglia have shown a role in initiating sexually dimorphic outcomes of NMO relapses compared to MS progression phases, as well as preferential AQP4-ab status, attack localization, and response to treatment however, researchers are uncertain of exactly how microglia are functionally reparative or damaging upon the release of trophic factors in NMO remission [223]. Overall, it is reasonable to note autoimmune diseases generally promote M1 microglial phenotypes and are more predominant in females, as summarized in Table 3. Whether females have a stronger pro-inflammatory response to AQP-4 positive antibodies and complement factors and the role of activated microglia in employing sex-differentiated profiles in the onset of disease awaits further investigation.

## 12. Worse Outcomes in Male CNS Infections Are Mediated by M2 Microglia

At first glance, the CNS would appear impenetrable to peripheral invaders. The cerebral endothelium forms the BBB, while epithelial cells of the choroid plexus form the blood–cerebrospinal fluid (CSF) barriers [15,225,226]. The brain is further enveloped by the surrounding avascular arachnoid epithelium under the dura mater [227]. Endothelial cells line the inside of blood vessels in close association to form tight cellular junctions crucial to microvessel maintenance, integrity, and permeability of the BBB. Additional neuroimmune interfaces with blood and neural tissue include the blood–spinal cord and blood-retinal barriers (BRB) [227,228]. Despite all of this, extracellular pathogens can cross the monolayer of tight junction-expressing endothelial or epithelial cells and cause infections by mechanisms categorized into transcellular, paracellular, and the Trojan horse methods.

Microbes can cross the BBB by gaining access to the luminal side of the blood vessel endothelium, where they can traverse through the endothelial cells by two mechanisms known as adsorptive-mediated transcytosis (AMT) and receptor-ligand mediated transcytosis (RMT) [229]. AMT relies on charge interactions rather than specific ligand-receptor binding between the microbe (ligand) and endothelial cell (receptor) to essentially absorb the infectious protein into the cell. In both transcellular and paracellular penetration, microbes adhere to brain microvascular endothelial cells; however, in paracellular penetration a microorganism may enter in between two tight junctions [229]. In contrast to direct movement of a microorganism across the BBB, microbial transfer occurs with the transmigration of an infected phagocyte. Infected leukocytes or other immune cells are taken advantage of for the invader to cross the BBB so a microorganism can gain CNS access [229]. Infiltrating, infected monocytes may be the primary carrier of human immunodeficiency virus-1 (HIV-1), well known to infect white blood cells using the C-C chemokine receptor type 5 (CCR5), CD4 receptor, and/or the C-X-C chemokine receptor type 4 (CXCR4) co-receptors [230]. Overall, it is important to note these microbe BBB routes and penetration strategies are not limited nor mutually exclusive such as in the case of *E. coli* K1 or streptococcal group B species [231,232,233,234,235].

A well-documented sex-differential disease susceptibility has been observed in clinical studies of the fungal pathogen *Cryptococcus neoformans* (Cn). Cn diseases across healthy and immunocompromised patients target males 2–3:1 compared to females, even prior to the beginning of the HIV epidemic [236,237,238,239,240]. Cn is a fungal species found worldwide, often associated with trees such as the mopane tree and in cultures from several types of bird excreta, particularly from pigeons [241]. A host can contract Cn by aerosol transmission, and likely while digging around soil or in humid environments [241]. In the United States, the incidence of cryptococcosis is 0.4–1.3 cases per every 100,000 and 2–7/100,000 for autoimmune deficiency syndrome (AIDS) patients [242]. An evidential SERO-epidemiology study showed the majority of child-derived samples from subjects living in the Bronx have been exposed to a Cn infection, with 70% of sera samples reacting to the immunoassays [243].

Additional considerations for the Cn pathogen rely on its opportunistic capacity and particular risk in HIV-positive or AIDS patients, older patients (ages 45 and above) demonstrating age-related immune system senescence, individuals with liver disease or diabetes, or cancer patients [244,245,246,247]. For immunocompromised individuals, this fungus can also promote the onset of chronic pneumonia, latent infection, nuchal rigidity or stiffness, or behavioral changes, followed ultimately by chronic meningitis upheld by uncontrollable fungal growth [246,248]. Although a depletion in CD4+ T cells is utilized as a diagnostic measure to initially determine infection (i.e., CD4 count less than 250), evidence of meningeal inflammation is often reported alongside substantially elevated levels of white blood cells such as lymphocytes in the CSF, referred to as lymphocytic pleocytosis. This is often seen in patients with late-stage HIV infections who have already begun antifungal therapy for the Cn infection and simultaneous antiretroviral therapy [247]. This rebounds with the flooding of CD4+ T cells into the brain due to severe infection as the immune system begins to recover and reawaken with antiretroviral therapy (ART). However, the opportunistic infection often triggers worse symptoms via an overexuberant immune response known as immune reconstitution inflammatory syndrome (IRIS) [249].

Cryptococcosis a systemic mycosis resulting from a highly angio-invasive hematogenous spread of Cn disseminating throughout the body often presenting as fatal cryptococcal meningitis or as a pulmonary or cutaneous cryptococcaemia infection [250]. Cn infection may also result in chronic meningoencephalitis, reported by a retrospective review article assessing medical records of all patients with a cryptococcus cryptococcosis between 2005 and 2017 in an inner-city medical center in the Bronx, NY [251]. Cn penetrates the brain parenchyma by a paracellular transversal over the BBB using plasmin or ammonia, or by Trojan horse mechanism with macrophages [252]. Cryptococcus exposure is often contained in healthy individuals and remains an asymptomatic infection, yet there have also been reports of a subclinical pneumonia upon a patient’s first contact with the organism [253]. In non-immunocompromised patients, the infection is often immediately isolated in the lymph nodes and the lungs; however, in some patients the yeasts’ spores can disseminate from primary inoculation sites and gravitate towards the CNS [254].

Clinical observations and experimental models depict sex hormones largely play a role in the underlying pathophysiological mechanisms leading to differential susceptibilities to various infections. For example, treatment of the Cn pathogen with exogenous testosterone significantly increases production of virulence factors relative to estrogen treatment [255]. Further, testosterone has demonstrated an immunosuppressive role in castrated male mice compared against intact females and males [256]. Experimental administration of testosterone into female mice generally increased their infectious susceptibility [256,257,258,259]. Immune resident phagocytosis in Cn hosts is reportedly enhanced in females compared to male counterparts in human and in rodent studies utilizing a synthetic estrogen compound [260]. On both opposing spectrums, higher endogenous levels of testosterone hormones in males, versus higher estrogens in females, may account for the opposing variances in Cn prevalence [261,262]. Healthy male Cn donor isolated macrophages demonstrated an increased splenic cell death rate in Balb/c strain mice; researchers have observed a significantly higher splenic fungal burden in males compared females [261]. In addition, different mouse strain susceptibilities to pulmonary Cn infection have been associated with pleiotropic differences in the immune response [262]. This paradigm is also apparent in bacterial infections of the CNS, as several studies have demonstrated estrogen as a supportive factor in regards towards an enhancing the immune response such as in experimental bacterial sepsis in rats [263,264]. In addition, 17β-estradiol has shown to protect ovariectomized female rodents in tissue lesion-inducing infections such as *H. Pylori* and *C. Burnetti* reducing bacterial loads and granulomatous responses in mice [265,266,267,268]. In combination with epidemiological insight, these data suggest Cn meningitis prevalence alongside the disproportionate incidence of infections in males over females is modulated by different genetic, hormonal, and experiential environments, as summarized in Table 4. The mechanistic processes underlying microglial modulation of a host’s sex-differentiated immune response await further elucidation.

### Mechanism of Microglial Involvement in Cryptococcal Meningitis

Researchers posit the complex Cn capsule plays a large role in its inadequate identification by peripheral immune cells due to its ability to conceal PAMPs on the cell surface [269]. Upon *Cryptococcus gatti* dissemination into the CNS, microglia can potentially recognize fungal pathogens with their TLRs, such as TLR-2 and 9, despite capsule masking [270,271,272]. Post-mortem autopsies and neurological assessments have demonstrated much of the Cn capsule is concentrated near and ingested by human microglia [269,273,274]. Recognition of PAMPs further initiate signal transduction cascades able to induce the canonical activation of MAPKs and NFkB, ultimately contributing to the destruction of pathogens as seen in peripheral phagocytic immune cell processes [275,276] Research suggests peripheral phagocytosis is activated by the Fc receptor for anti-capsular immunoglobulin G (IgG); however this remains a topic in need of further elucidation for Cn species [277]. This monoclonal antibody immune complex can further promote the release of pro-inflammatory cytokines IL-1β, IL-4, IL-6, IL-10, IL-12, IL-23, as well as TNFα, and also upregulation of MHC II markers and CD11c integrin [278,279]. The overall phagocytic capacity and expression of proinflammatory molecules relevant to the clearance of cellular debris from microglia is downregulated by Cn growth [276,280]. While phagocytosis is a fundamentally primary immune response, increased CD4+ presence, MHC II constituents, cytokines such as IL-12 and IL-23p19, and increased iNOS expression have all mechanistically presented as enriching the response network comprising of cell-to-cell crosstalk in the rodent PNS and CNS [281,282].

Conceptually, the molecular mechanisms of microglial PAMP recognition of Cn and other fungi are studied from the outer periphery into the CNS. Upon exposure to Cn antigens, microglia can recruit peripheral macrophages and CD4+ or CD8+ cells to further respond to local infections, without which the host quickly succumbs to disease [283,284,285]. Further, myeloid dendritic antigen presenting cells (APCs) coordinate CD4+ and CD8+ T cell functions in anti-tumor immunity models and may additionally promote microglial activation in cryptococcal meningitis [286].When stimulated, microglia fundamentally attempt to achieve a homeostatic environment alongside other immune cell types. Particularly relevant to sexual dimorphism, the synchronized and sex-typical behavior of microglia regarding the immune response may provide relevant insight for future therapeutic perspectives. For example, Cn-infected mice promote the microglial production of M2-associated CCL2, increasing disease onset and susceptibility [287]. Further research has reported a correlation between low IFNγ levels and late stages of Cn infection [288]. Current data on mechanisms of host resistance such as complement system activation and similar involved pathways are limited; however, opsonization is a critical step in allowing microglia to efficiently consume Cn cells [289].

Fundamental female predominance of M1 phenotypes tend to resolve infection more efficiently, whereas male lack of M1-phenotypes toward a more M2-male microglial approach may underlie the onset of chronic meningitis. Moreover, intrinsic sex differentiated microglial phenotypes may provide an explanation for differential infectious disease outcomes and progression, as summarized in Table 5. Overall, with respect to sexual trajectories and hormone-involved processes pertaining to sexual outcomes, numerous studies show that females generally demonstrate an enhanced infection clearance capacity compared to males despite age [290,291,292]. This capacity may pertain to M1 predominant microglial-modulatory phenotypes relative to epidemiological sex-differential data. M2 predominance of microglia alongside lifelong endogenous circulation of testosterone in males may function as combined susceptibility factors advantageous toward virulent penetration, onset, progression, and worsening of symptoms in Cn infections, while M1-driven female microglial responses would promote viral clearance and recovery [293].

## 13. Male Predominance of Glioblastoma Multiforme Is Mediated by M2 Microglia

Primary CNS tumors are heterogeneous and can range from benign to malignant. Tumors of neuroglia (gliomas) are the most common malignant cancerous tumor arising in the CNS [294]. Unlike most cancers characterized by stages, the classification system of glioma assesses how aggressive the tumor appears by grades I-IV under a microscope [295]. Upon the diagnosis of a CNS cancer, a clinician may comparatively analyze radiological images, pathological images, and molecular profiles to ultimately finalize an individual treatment methodology tailored to patient specificities such as age and sex [296,297]. Glioma classification assesses the degree of proliferation in concert with the presence or absence of necrosis, mitotic cell index severity, and genetic or molecular markers [298]. Tumors arising from astrocytes generally tend to form in the cerebral cortex or the corpus callosum yet may be found anywhere in the brain or spinal cord [299,300,301].

Grade I gliomas or juvenile pilocytic astrocytomas are benign and are often resolvable with surgical treatment alone [302]. Grade II or low-grade gliomas can recur over time and most commonly require additional treatment post-surgery. Recurring conditions can transform the tumor into more pathologically aggressive form known as grade III or high-grade anaplastic gliomas [303]. Grade III glioma treatment post-surgery is often followed by postoperative adjuvant treatments such as chemotherapy and radiation in patients where it is tolerable [304]. Grade IV gliomas or glioblastoma multiforme (GBM) are the most aggressive tumors, and often demonstrate a cloud-like cell growth pattern that makes it difficult to visually differentiate by surgeons due to the inability to decipher where the tumor ends and also due to a combination of small tendril-like projections and severe swelling thought to harbor cancerous microscopic cell particulates remaining from tumorous masses [305,306]. GBM’s aggressively irregular nature also reportedly disrupts BBB homeostasis, promoting extensive edema further supporting high-grade malignancy and infiltration of peripheral macrophages [307]. The GBM tumor presents with cystic and necrotic areas, as well as microvascular proliferation [308]. Notably, surgery alone has proved to be an inefficient curative treatment modality for GBM; therefore, follow-up therapies are necessary to address recurrent, intractable disease.

Interestingly, a greater incidence and worse outcome of GBM is consistently reported in several studies of male comparisons to females [309,310,311]. GBM incidence is 1.6 times higher in men than women, as opposed to similar low-grade glioma incidence amongst both sexes [312]. This holds true for both primary and recurrent forms of GBM [313,314]. GBM predominance in males is also recognized amongst young children [315,316]. Regardless of ovariectomy, testes castration, treatment, or age-related decline, GBM tumors are more frequently prevalent in males, evident across human studies and animal models [317,318,319]. This disparity does not only affect the rate of diagnosis, as females also demonstrate substantially increased survival rates compared to males [319,320].

Sex-specific effects in GBM tumorigenesis may be due to intrinsic cellular differences or circulating endogenous hormones. For example, orchiectomized male rats notably develop fewer glial tumors and demonstrate a greater survival outcome than intact males [321]. The higher incidence of GBM in adult males has been associated with the upregulation and increased expression of androgen receptors, which are linked to increased GBM proliferation [322]. Silencing of these receptors has become a promising therapeutic target as this increases GBM cell death both in vitro and in vivo [323,324]. Beyond directly impacting cancer proliferation itself, administration of testosterone increased permeability of the blood-brain barrier and promoted GBM migration and invasion [325,326]. Estrogen exposure during GBM may have a protective effect, as female rats implanted with U87MG cells survived longer than males, and ovariectomy negated this advantage [327]. However, women post-menopause still demonstrated an increased survival rate compared to younger females and to older females after menopause, respectively [328,329]. Beyond sex hormones, female GBM cells show greater susceptibility to radiation compared to males, potentially through increased p21-induced irradiation related cellular senescence [330]. Taken together, it is clear that GBM is a male-dominant disease throughout all age groups, as shown in Table 6.

### Mechanisms of Microglial Involvement in GBM

As previously mentioned, microglia are resident immune cells responding to local signals within the brain, thereby implying their innate residence under tumor-threatening CNS conditions. Glioma associated microglia and macrophages (GAMs) have often been coupled in studies seeking to ascribe their roles in tumor invasion, migration, and escape from anticancer therapeutics [332,333]. Through these studies, evidence has shown that GAMs are crucial to the development, maintenance, and eventual eradication of cancer cells in the tumor microenvironment. GAMs can functionally reprogram the tumor microenvironment to promote proliferation, metastasis, and processes fostering invasion of healthy tissue [334]. Microglial cells have a double-edged function in modulating GBM progression and recovery. On one hand, M1 macrophages function by stimulating Th1 cell responses, as well as scavenging and destroying tumor cells through the process of phagocytosis [335]. However, most GAMs are thought to resemble anti-inflammatory M2 phenotypes activated by cytokines such as IL-4, IL-10, and IL-13 [6,336]. The presence of these cells in the tumor microenvironment is favorable to GBM growth and thus correlates to poor disease prognosis and diminished response to radiotherapy [337,338]. Furthermore, M2 macrophages express more angiogenic factors and proteases, promoting tumor growth and metastasis, compared to their M1-polarized counterparts [6]. The precise timing in which microglia switch to the tumor-supportive phenotype during the progression of GBM is currently unknown; however, cytokines produced by cancer cells in the tumor microenvironment have been shown to suppress innate microglial immune activation processes, preventing anti-tumorigenic activity [339,340]. A study on chimeric mice demonstrated monocyte-derived macrophages infiltrate the brain solely in the late progression stages of gliomas at about 21 days after glioma tumor engraftment, and that microglia predominate at the tumor site prior to that time [341]. Further, cytokine profiling studies revealed pro-inflammatory factors such as IL-8 were expressed by naïve microglia, and not by tumor-associated microglia, further demonstrating a pathogenic M2 phenotype preference by glioma-adjacent microenvironments [340,342]. The presence of these anti-inflammatory M2 GAMs further supports the development of an immunosuppressive, pro-tumorigenic milieu [343].

Manipulation of the balance between M2 and M1 microglial populations shows great potential as a therapeutic target for GBM. Neuropilin-1 (Nrp1) signaling in microglia and macrophages promotes an M2 shift. If this receptor is blocked either through genetic knockout or pharmacological disruption, these cells instead shift towards an M1 phenotype [344]. In animal models of GBM, Nrp1 disruption promoted the generation of anti-tumorigenic immune responses that showed significant reduction in tumor size and increased survival [345,346]. Based on these data, it is clear that M2 GAMs are supportive of GBM growth and promote the development of more severe disease, as seen in Table 7. Given the predominance of M2 microglia in males relative to females, their increased risk of developing GBM could potentially be explained by this neuroimmune disparity.

## 14. Conclusions

As discussed above, microglial phenotypes diverge substantially between males and females. These disparities begin to surface at birth, where developmental trajectories are modulated by the presence of XX or XY chromosomes. The onset of endogenous programming for microglial phenotypes in early life experientially shape differential immune approaches over time. Early male-programmed microglial hyperactivation rapidly tapers into a conceptually more immature immune response. In comparison, default feminized immune profiles are associated with greater inflammation later in life, in part due to considerable hormonal profile alterations and age-related senescence. Due to the microglial ability to maintain sexual phenotypes over one’s lifetime or when transplanted into the opposite sex, these cells maintain control of an individual’s disease response. Increasing evidence supports the concept that males develop a more immunosuppressive, neuroprotective M2 microglial phenotype, while in contrast female microglia take on an M1-predominant inflammatory functionality. The exact mechanism behind these differences is incompletely understood but may in part be due to hormonal influences as testosterone has been linked to reduced microglial activity and anti-inflammatory cytokine production.

As microglia are inexorably linked to the pathology of CNS disease, their sexually dimorphic function becomes increasingly relevant to understanding the progression and treatment of these disorders. Autoimmune disease, glioblastoma, and infection have all shown disparate rates of diagnosis between men and women, with greater numbers of female patients afflicted with MS and NMO, and more prevalent diagnoses of GBM and *Cryptococcus neoformans* infection in concert with worse outcomes in males. What is critical is that autoimmunity in the CNS is primarily mediated by aberrant excessive activation of M1 microglia, while abundance of M2 microglia is deleterious when facing cancer or infection (Figure 2). Taken together, these data indicate that the reason why women are more likely to suffer from MS may be that they are predisposed to developing an inflammatory microglial response, while men are primed towards anti-inflammatory immunosuppressive M2 responses that allow cancer and infections to thrive.

Moving forward, the question remains how to use this information to better support medical advancement. Manipulation of microglial functionality is a promising avenue to treat a wide variety of CNS disorders [348]. However, it is important to note that all microglia are not created equal, as a hypothetical therapeutic that promotes M1 responses to treat GBM could potentially be very effective in males but result in immune overactivation in females. It is thus vital to not only include men and women in trials for drugs in development, but also to account for sexually dimorphic immune responses in determining how best to approach patient-specific therapies. Understanding the role of microglia in mediating these divergent disease outcomes could bring about a new paradigm of tailored interventions.

## Figures and Tables

**Figure 1 ijms-24-04739-f001:**
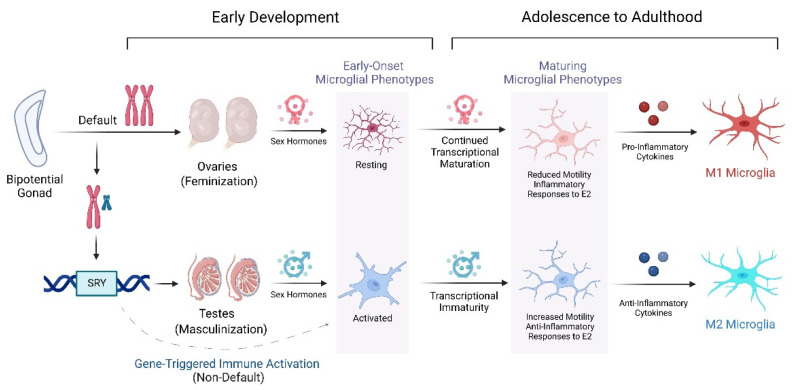
Sexually dimorphic microglial development. Default developmental programming that results in producing ovaries is altered by the presence of the SRY gene on the Y chromosome, producing testes. Sex hormones generated from these gonads (estrogen from ovaries, testosterone from testes) shape initial microglial phenotypes, with increased activation seen in males. Developmental maturation shortly downregulates in male microglia, ultimately assuming a more transcriptionally immature profile compared to female microglia that continue to mature. Cytokine production and differential responses to estradiol (E2) dictate an overall M1 predominance in females and M2 predominance in males. (Created with BioRender.com on 6 February 2023).

**Figure 2 ijms-24-04739-f002:**
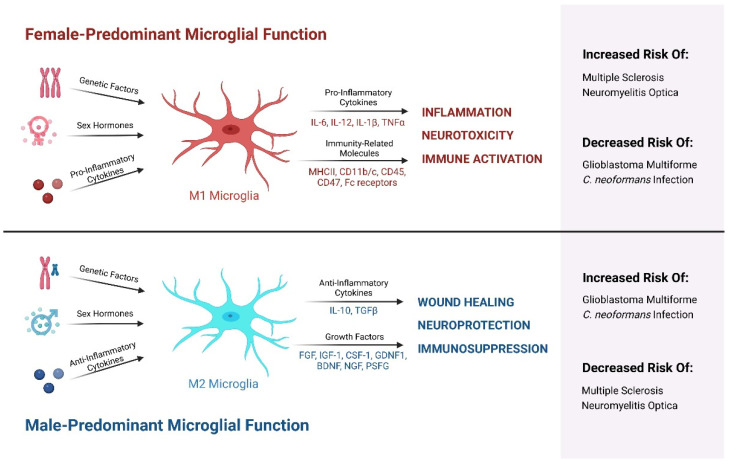
Sexually dimorphic pathological activation of microglia. Genetic and environmental factors such as sex hormones and cytokines shift microglia to a female-predominant M1 and male-predominant M2 phenotype. The intrinsic functions of these sexually dimorphic cells then support the development or resolution of a variety of CNS pathologies. (Created with Biorender.com on 6 February 2023) Abbreviations: IL-6, interleukin-6; IL-12, interleukin-12; IL-1β, interleukin-1 beta; TNFα, tumor necrosis factor alpha, MHCII, major histocompatibility complex II; IL-10, interleukin-10; TGFβ, transforming growth factor beta; FGF, fibroblast growth factor; IGF-1, insulin-like growth factor-1; GDNF1, glial-derived neurotrophic factor 1; BDNF, brain derived neurotrophic factor; NGF, nerve growth factor; PSFG, pro-survival factor granulin.

**Table 1 ijms-24-04739-t001:** Microglial cells are predominantly M1 in females, and predominantly M2 in males.

Model	Species	Evidence Supporting Female M1 Predominance	Reference
Immunohistochemistry for Iba1	Rat	Increased numbers of activated microglia in P60 females	[115]
Whole transcriptome profiling of primary microglial cells	Mouse	Female microglia have a more transcriptionally mature profile, increased expression of inflammatory transcripts in adulthood	[119]
Whole transcriptome profiling of primary microglial cells	Mouse	Increase expression of MHC I, complement pathway, and inflammatory genes in aged females	[120]
Transwell migration assays	Rat	Increased motility in male microglia	[122]
Renal ischemia/reperfusion	Rat	Increased IL-10 production following testosterone treatment	[128]
Western blotting	Mouse	Increased expression of inflammation-associated CCR5 following estrogen treatment	[133]
RT-PCR	Rat	E2 diminished the inflammatory release of IL-1β following LPS stimulation in male microglia, but enhanced the inflammatory response in female microglia.	[134]
RNASeq	Mouse	Microglia from female mice had higher mRNA levels for TNFα, IL-1β, and IL-6 than those from males	[137]
Brain stab wound model, MHC II immunostaining	Rat	Decreased microglial reactivity with testosterone treatment following brain injury	[126]
IL6 stimulation	Rat	Diminished stress responses with reduced corticosterone and ACTH release in rats with increased testosterone	[138]
Whole blood samples	Human	Higher endogenous testosterone associated with lower serum cytokine levels	[139]
Various clinical studies of human disease	Human	Suppressed TNFα, IL-6, IL-1β and increased IL-10 with high levels of testosterone	[129,140]

**Table 2 ijms-24-04739-t002:** MS and NMO are female predominant.

Disease	Species	Evidence Supporting Female Predominance	Reference
Multiple Sclerosis	Human	The female to male ratio was reported as 2.35:1.	[151]
Multiple Sclerosis	Human	Female incidence of MS in all races is about three times of their male counterparts	[152]
Multiple Sclerosis	Human	3:1 ratio of MS diagnosis between females and males	[169]
Multiple Sclerosis	Human	Women with MS had more inflammatory MS lesions than men.	[157]
Neuromyelitis Optica Spectrum Disorders	Human	Female patients were more likely to be positive for susceptibility-associated AQP4-abs (92% vs. 55%; *p* < 0.001)	[168]
Neuromyelitis Optica Spectrum Disorders	Human	4.6:1 ratio between females and males for NMOSD diagnosis; among AQP4-ab seropositive groups the ratio is 12:1	[170]
Neuromyelitis Optica Spectrum Disorders	Human	High AQP4-antibody seropositivity prevalence in females compared to males (9:1)	[171]
Neuromyelitis Optica Spectrum Disorders	Human	Seropositivity for AQP4-IgG is 8:1 between females and males, and particularly in people ages 18 years and older.	[172]
Neuromyelitis Optica Spectrum Disorders	Human	Reduced early-onset optic neuritis in male patients	[173]

**Table 3 ijms-24-04739-t003:** M1 microglia drive CNS autoimmune pathology.

Model	Species	Evidence Supporting M1-Associated Pathology	Reference
EAE	Mouse	Microglial transcriptomes show increased neuroinflammatory and immune signaling pathways during EAE progression	[175]
EAE	Mouse	Polarization of microglial cells to M2 substantially reduces EAE symptoms and progression	[181,182,183]
EAE	Mouse	Ablation of microglia improves EAE symptoms	[184]
EAE	Mouse	Increased inflammatory cytokine production and Th17 activation in microglial repopulated mice during EAE	[185]
EAE	Mouse	Deletion of M1-associated MAPK pathway proteins in microglia is protective in EAE.	[194]
EAE–optic neuritis (EAEON)	Mouse	TNFα was significantly upregulated in the optic nerve of EAE animals with a peak at 35 dpi.	[224]
Lysolecithin-mediated demyelination, primary microglial culture	Mouse	Microglial cells switch from M1 to M2 phenotypes at the initiation of remyelination, M2 cells promote oligodendrocyte differentiation.	[179]
EAE	Mouse	Infiltrating T cells promote inflammatory microglial activation, which recruit additional peripheral immune cells	[186]
Postmortem MS autopsy samples	Human	Increased myelin phagocytosis and T cell activation by microglia in areas of active inflammation	[187]
Postmortem MS autopsy samples	Human	Active and chronic active MS lesions showed abundant expression of M1 markers, with M2 markers lacking overall	[196]
Injection of patient-derived NMO-IgG	Rat	High levels of IL-1β positive microglia found in active NMO lesions	[210]
Primary microglial culture	Rat	NMO-IgG promotes release of M1-polarizing, inflammatory CCL5 and C3 from astrocytes	[212]
Infusion of patient-derived serum NMO-IgG or AQP4-IgG	Mouse	Direct interaction between astrocyte and microglial soma following antibody infusion. Increased inflammatory microglial activation and C1q production in NMO lesions, elimination of NMO-IgG motor impairment following microglial ablation	[199]

**Table 4 ijms-24-04739-t004:** Males are more susceptible to infections than females.

Disease	Species	Evidence Supporting Female Predominance	Reference
*Cryptococcus neoformans* infection	Human	2:1 ratio between male and female patients prior to the HIV epidemic	[236]
*Cryptococcus neoformans* infection	Human	3:1 ratio between male and female HIV negative patients, 8:1 ratio between male and female HIV positive patients	[238]
*Cryptococcus neoformans* infection	Human	7:3 ratio between male and female patients	[239]
*Cryptococcus neoformans* infection	Human	97.3% of HIV+ Cn patients were male; 62.7% of HIV- Cn patients were male	[240]
*Candida albicans* infection	Mouse	Testosterone functions in an immunosuppressive manner, infusion of testosterone into female mice increases infections	[256]
*Nippostrongylus brasiliensis* infection	Mouse	Parasite infection commonly recurred in male animals and rarely in females; this recurrence in males was eliminated following orchiectomy	[258]
*Cryptococcus neoformans* infection	*In vitro*	Synthetic estrogen compounds have potent antifungal activity	[260]
*Cryptococcus neoformans* infection	Human	High levels of testosterone were associated with greater Cn pathology, male-derived strains had shorter doubling times, greater fungal burden and increased splenic cell death rate	[261]
*Vibrio vulnificus* infection	Rat	Gonadectomy in females increases mortality; estrogen treatment decreases mortality in males and females	[263]
*Enterococcus faecalis* infection	Rat	Estradiol treatment reduces bacterial counts	[264]
*Helicobacter pylori* infection	Mouse	Reduced gastric lesions in estradiol treated mice	[265]
*Coxiella burnetti* infection	Mouse	Bacterial load and granuloma numbers were reduced in female mice, estradiol treatment in ovariectomized mice had an analogous effect	[268]

**Table 5 ijms-24-04739-t005:** M1 inflammatory responses promote recovery from infection.

Model	Species	Evidence Supporting M2-Associated Pathology	References
Infection with *Cryptococcus neoformans*	Mouse	Inflammatory IFNγ is critical for microglial anticryptococcal efficacy	[278]
Infection with *Cryptococcus neoformans*	Mouse	Microglial proliferation is suppressed by Cn infection, reduction of IFNγ and delayed microglial activation associated with worse disease symptoms	[276]
Infection with *Cryptococcus neoformans*	Mouse	Increased microglial numbers in the brains of immune mice	[283]
Microglial cell culture	Human	Increased M1-associated MIP-1α, MCP1, and IL-8 in the presence of opsonized Cn	[284]
Infection with *Cryptococcus neoformans*	Mouse	M2-associated CCL2 production correlates to increased Cn susceptibility	[287]
* Cryptococcus gatti * infection	Mouse	Loss of M1-associated TLR9 results in a higher mortality rate and a greater number of fungal CFUs in the brain. Reduced inflammatory IFNγ and IL-17 cytokines were correlated with higher fungal burden.	[272]
Intracerebral *Cryptococcus gatti* infection	Mouse	Administration of M1-polarizing IL-2 activated microglia and significantly prolonged survival time.	[278]
Analysis of neutralizing antibody responses to the trivalent seasonal influenza vaccine	Human	Female serum samples showed an elevated expression of inflammatory cytokines, STAT3 proteins in monocytes, and antibody responses to the vaccine compared to males	[290]

**Table 6 ijms-24-04739-t006:** GBM is a male-predominant disease.

Model	Species	Evidence Supporting Male Predominance	References
Central Brain Tumor Registry of the United States, 2004–2016	Human	57.5% of GBM patients were male, higher risk of death in male patients	[310]
Multi- institutional repository of clinical data from 1,400 GBM patients	Human	60.5% of patients were male	[311]
Incidence of primary brain and other CNS tumors in the US	Human	Glioblastoma is 1.57 times more common in males.	[312]
SEER database from 2000–2010	Human	Of 14,675 GBM patients, 59.4% were male.	[313]
GBM tumor samples from the Chinese Glioma Genome Atlas	Human	Male to female ratio of primary GBM is 1.67, recurrent GBM is 1.75.	[314]
Retrospective Analysis of all patients ≤21 years with high-grade glioma	Human	Female patients demonstrated enhanced survival following gross total resection of tumors compared to male patients, with a mean of 8.1 years in females and 2.4 years in males. Females had increased progression-free survival.	[316]
Multivariate Analysis of the Project of Emilia Romagna on Neuro-Oncology (PERNO) registry	Human	58.6% of patients were male. In those that had hypermethylated O6 Methylguanine-DNA-Methyltransferase (MGMT), methylated females had longer survival compared to methylated males.	[317]
Analysis of serial Magnetic Resonance Images	Human	Quantitative imaging-based measure of response revealed standard therapy is more effective in female GBM patients.	[318]
SEER database from 2000–2008	Human	5-year cancer-specific survival rate was 6.8% in males and 8.3% in females	[320]
OBTS database from 2007–2017	Human	Male median survival was 15.9 months post-diagnosis, 22.6 months in females.	[319]
3,4- benzopyrene induced glial tumor formation	Rat	77.8% of intact male rats developed glial tumors, which dropped to 50% following castration	[321]
GBM cell lines	Human	Upregulation of androgen receptor in all eight examined GBM cell lines, DHT administration prevents TGF β -mediated inhibition of GBM.	[322]
Glioma patients, cell lines	Human	Elevated androgen receptor and serum testosterone in GBM patients relative to healthy controls, androgen receptor antagonists inhibit proliferation.	[323]
GBM female patient samples	Human	Increased androgen receptor mRNA in 93% of samples	[331]
GBM patient specimens, cell lines	Human	Increased androgen receptor protein in 56% of GBM samples, 30% are constitutively active. Androgen receptor siRNA silencing induced GBM cell death.	[324]
GBM cell culture	Human	Testosterone increased GBM proliferation, migration, and invasion.	[326]
Intracerebral implantation of cultured U87MG cells	Rat	The apoptotic index for U87 tumor cells was significantly higher in females than males. Females survived longer than males and ovariectomy negated the advantage.	[327]
Incidence Rate Calculation from New York State Cancer Registry Data	Human	The female population had a lower risk of developing GBM than males, with the protective effect of sex most evident at the approximate age of menopause. Protective effects decreased in postmenopausal age strata.	[328]
Histologically confirmed cases of glioma identified from population- based cancer registries	Human	Risk of glioma increased among women with a relatively older age at menarche and among women with a history of hysterectomy.	[329]
Irradiation of mouse astrocytes	Mouse	Significantly decreased proliferation in female astrocytes compared to male post-irradiation, increased markers of cellular senescence in females	[330]

**Table 7 ijms-24-04739-t007:** GBM pathology is mediated by M2 GAMs.

Model	Species	Evidence Supporting Male M2 Predominance	References
Primary glioma patient samples	Human	The amount of M2 GAMs increases with malignancy grade and is associated with shorter survival.	[336]
Patient-derived glioma stem cells	Human	Short-term relapse following radiation therapy was correlated with high M2 GAM frequency. The frequency of M2 macrophage/microglia was increased in recurrent mesenchymal GBM compared to primary non-mesenchymal GBM.	[337]
Mouse- and patient-derived cells	Mouse, Human	Microglial/macrophage activation by amphotericin B upregulates the M1 markers MCP-1 and IL-8, and inhibits brain tumor initiating cells (BTICs). Cells cultured from the tumor microenvironment could not inhibit BTICs	[339]
Chimeric mice to track origins of GAM populations	Mouse	Mouse chimeras showed that macrophages infiltrated at late stage GBM, and comprise only ¼ of the total myeloid-cell fraction in experimental gliomas. Microglia were found to up-regulate CD45 expression levels and represent up to 40% of the CD45^high^ cell population.	[340]
Genome sequencing profile from the Chinese Glioma Genome Atlas (CGGA) database	Human	In analysis of human GBM genomes, the high-risk group was characterized by upregulated mRNA expression of M2 microglia/macrophage markers and higher levels of IL-10 and TGFβ1.	[341]
Transcriptomic profile analysis of experimental C6 gliomas	Rat	In microglia from rat gliomas, genes characteristic of M1 activation such as *Tlr2*, *Tlr4* and *Il18*, *Cd80*, *Il12a*, *Il15* were downregulated, and gene expression characteristic for M2 activation such as *Tgm2*, *Il1rn*, *Cxcl10*, *Cxcl16*, *Ccl2*, *Ccl17*, *Ccl22*, *Vegfa*, *Stat3*, *Arg1* and *Mmp14* were upregulated.	[342]
Implantation of GL261 GBM cells	Mouse	Pharmacological blockade or conditional knockout of Nrp1 from microglia/macrophages polarizes GAMs towards an M1 phenotype, reduces tumor size and vascularity and increases survival	[345]
Implantation of GL261 GBM cells in chimeric mice	Mouse	Conditional knockout of Nrp1 from microglia/macrophages polarizes GAMs towards an M1 phenotype and increases antitumor cytotoxic T cell immune responses	[346]
GBM tissue specimens of from the Neurosurgery Unit of the Catholic University Medical School	Human	Increased M2-specific CD163 expression was observed within the tumor than in the surrounding periphery.	[347]

## Data Availability

Not applicable.

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
