# Peer review of "The Pathological Activation of Microglia Is Modulated by Sexually Dimorphic Pathways"

_ijms, 2023, doi:10.3390/ijms24054739_

Round 1
Reviewer 1 Report
I thank the authors for the great work done in the field of literature analysis on the issues under study. The title of the manuscript fully reflects the essence of the study.
I have a few minor comments that do nothing to diminish the value and quality of the manuscript:
1. In the introduction chapter, a lot of disparate data is described. I would advise the authors to work through this chapter and make it more comfortable to read and understand.
2. I also recommend authors to provide a schematic illustration of the differences in microglial response depending on sex in order to visualize the mechanisms described in the text.
Author Response
We thank the reviewer for their time in reviewing our paper and for their helpful comments. Our revisions are as follows:
Point 1. In the introduction chapter, a lot of disparate data is described. I would advise the authors to work through this chapter and make it more comfortable to read and understand.
We have rewritten the introduction to improve clarity and to connect it more effectively to the content of the review.
Point 2. I also recommend authors to provide a schematic illustration of the differences in microglial response depending on sex in order to visualize the mechanisms described in the text.
We have added three figures to the paper - a graphical abstract, an overview of microglial development, and a concluding figure summarizing sexually dimorphic microglial function in pathological contexts.
Reviewer 2 Report
Reviewer comments and suggestions
This review discusses the developmental and environmental indications that support microglial polarization towards different phenotypes and mentioning sexually dimorphic factors that can influence this process. The authors also describe a variety of CNS disorders including autoimmune disease, infection, and cancer that demonstrate disparities in disease severity or diagnosis rates between males and females and speculate that microglial sexual dimorphism underlies these differences.
Comments
- The section Title (DIVERSE MICROGLIAL MORPHOLOGIES IN NEURODEGENERATIVE DISEASE ) seems to be exaggerated after going through the paragraphs the authors did not explain all neurodegenerative diseases, they should be specific for the section and included text in all sections.
- Comments for section 4th paragraph ( SEXUALLY DIMORPHIC DEVELOPMENT OF THE NEUROIMMUNE SYSTEM) The author needs to mention the references which recent studies they were discussing at the end of the sentences
- I have observed that the authors added many tables, if they add figures instead of explaining it would be better for a common reader of your paper. Please try to add up at least 2 figures on microglial sexual dimorphism.
- In my view, the authors need to delete the references present in the conclusion sections as it might be already been discussed in the text of the manuscript.
- The reference style needs to be modified based on mdpi guidelines.
Author Response
We thank the reviewer for their time and their helpful comments on our paper. Our revisions are described below:
Point 1. The section Title (DIVERSE MICROGLIAL MORPHOLOGIES IN NEURODEGENERATIVE DISEASE ) seems to be exaggerated after going through the paragraphs the authors did not explain all neurodegenerative diseases, they should be specific for the section and included text in all sections.
This section has been renamed "Microglia assume diverse morphologies in disease or injury" to more accurately represent the included text.
Point 2. Comments for section 4th paragraph ( SEXUALLY DIMORPHIC DEVELOPMENT OF THE NEUROIMMUNE SYSTEM) The author needs to mention the references which recent studies they were discussing at the end of the sentences
The references for this line have been added as requested.
Point 3. I have observed that the authors added many tables, if they add figures instead of explaining it would be better for a common reader of your paper. Please try to add up at least 2 figures on microglial sexual dimorphism.
Three figures have been added to this manuscript - a graphical abstract, a summary of sexually dimorphic microglial development, and a summary of microglial sexual dimorphism in pathological contexts.
Point 4. In my view, the authors need to delete the references present in the conclusion sections as it might be already been discussed in the text of the manuscript.
All of the references from the conclusions section excluding one that focused on potential future avenues of research were removed. This reference was not discussed earlier in the paper.
Point 5. The reference style needs to be modified based on mdpi guidelines.
The reference style has been corrected.